# Apicomplexan-like parasites are polyphyletic and widely but selectively dependent on cryptic plastid organelles

Jan Janouškovec[1]*, Gita G Paskerova[2], Tatiana S Miroliubova[2,3], Kirill V Mikhailov[4,5], Thomas Birley[1], Vladimir V Aleoshin[4,5], Timur G Simdyanov[6]

[1]Department of Genetics, Evolution and Environment, University College London, London, United Kingdom; [2]Department of Invertebrate Zoology, Faculty of Biology, Saint Petersburg State University, St. Petersburg, Russian Federation; [3]Severtsov Institute of Ecology and Evolution, Russian Academy of Sciences, Moscow, Russian Federation; [4]Belozersky Institute for Physico-Chemical Biology, Lomonosov Moscow State University, Moscow, Russian Federation; [5]Kharkevich Institute for Information Transmission Problems, Russian Academy of Sciences, Moscow, Russian Federation; [6]Faculty of Biology, Lomonosov Moscow State University, Moscow, Russian Federation

**Abstract** The phylum Apicomplexa comprises human pathogens such as *Plasmodium* but is also an under-explored hotspot of evolutionary diversity central to understanding the origins of parasitism and non-photosynthetic plastids. We generated single-cell transcriptomes for all major apicomplexan groups lacking large-scale sequence data. Phylogenetic analysis reveals that apicomplexan-like parasites are polyphyletic and their similar morphologies emerged convergently at least three times. Gregarines and eugregarines are monophyletic, against most expectations, and rhytidocystids and *Eleutheroschizon* are sister lineages to medically important taxa. Although previously unrecognized, plastids in deep-branching apicomplexans are common, and they contain some of the most divergent and AT-rich genomes ever found. In eugregarines, however, plastids are either abnormally reduced or absent, thus increasing known plastid losses in eukaryotes from two to four. Environmental sequences of ten novel plastid lineages and structural innovations in plastid proteins confirm that plastids in apicomplexans and their relatives are widespread and share a common, photosynthetic origin.
DOI: https://doi.org/10.7554/eLife.49662.001

*For correspondence: janjan.cz@gmail.com

**Competing interests:** The authors declare that no competing interests exist.

## Introduction

The phylum Apicomplexa is a major group of protistan parasites important in animal disease globally (we will use the name Apicomplexa hereinafter for the clade of parasites sensu stricto; a synonym of the taxon Sporozoa Leuckart, 1879; *Adl et al., 2019*). The group includes the human pathogens *Plasmodium* (haemosporidians), *Toxoplasma* (eucoccidians), *Babesia* (piroplasms), and *Cryptosporidium* (cryptosporidians), whose cell biology and genomes have been extensively studied. Conversely, most apicomplexans from invertebrates such as eugregarines, archigregarines, blastogregarines, protococcidians, and agamococcidians lack detailed genomic information, in part because they cannot be cultured in laboratory conditions. Because these uncultured groups are also deep-branching lineages with unresolved relationships to the medically important taxa (*Leander and Ramey, 2006*; *Rueckert et al., 2011*; *Simdyanov et al., 2018*; *Simdyanov et al., 2017*), the lack of their genomes and cell biology data hinders our understanding of apicomplexan evolution and the origin of

**eLife digest** Microscopic parasites known collectively as apicomplexans are responsible for several infectious diseases in humans including malaria and toxoplasmosis. The cells of the malaria parasite and many other apicomplexans contain compartments known as cryptic chloroplasts that produce molecules the parasites need to survive. Cryptic chloroplasts are similar to the chloroplasts found in plant cells, but unlike plants the compartments in apicomplexans are unable to harvest energy from sunlight.

Since the cells of humans and other animals do not contain chloroplasts, cryptic chloroplasts are a potential target for new drugs to treat diseases caused by apicomplexans. However, it remains unclear how widespread cryptic chloroplasts are in these parasites, largely because few apicomplexans have been successfully grown in the laboratory.

To address this question, Janouškovec et al. used an approach called single-cell transcriptomics to study ten different apicomplexans. This provided new data about the genetic make-up of each parasite that the team analysed to find out how they are related to one another. The analysis revealed that, unexpectedly, apicomplexan parasites do not share a close common ancestor and are therefore not a natural grouping from an evolutionary perspective. Instead, their similar physical appearances and lifestyles evolved independently on at least three separate occasions.

Further analysis demonstrated that cryptic chloroplasts are common in apicomplexan parasites, including in lineages where they were not previously known to exist. However, at least three lineages of apicomplexans have independently lost their cryptic chloroplasts.

The findings of Janouškovec et al. shed new light on the importance of chloroplasts in the evolution of life and may help develop new treatments for diseases caused by apicomplexan parasites. Several drugs targeting the cryptic chloroplasts in malaria parasites are currently in clinical trials, and this work suggests that these drugs may also have the potential to be used against other apicomplexan parasites in the future.

DOI: https://doi.org/10.7554/eLife.49662.002

parasitism itself. This, in turn, limits insights into infection mechanisms across the group: the structural and molecular make-up of machineries for host cell invasion such as the apical complex, pellicle, and glideosome, and how these relate to parasite life cycles, host preferences, and habitats.

Apicomplexans are evolutionarily derived from mixotrophic algae, a realization that first came with sequencing a plastid genome in *Plasmodium* and localizing it into a cryptic, non-photosynthetic organelle called 'the apicoplast' (*Gardner et al., 1991*; *McFadden et al., 1996*; *Wilson et al., 1996*). The apicoplast is a four-membrane plastid (a broader term we will use hereinafter to describe the organelle in both parasitic and free-living organisms), which is derived from a secondary endosymbiont. Where exactly this plastid endosymbiont came from was settled by data from two newly discovered photosynthetic relatives of Apicomplexa, *Chromera velia* and *Vitrella brassicaformis* (*Janouskovec et al., 2010*; *Moore et al., 2008*; *Oborník et al., 2012*). The photosynthetic plastids in *Chromera* and *Vitrella* are also surrounded by four membranes and they share conspicuous similarities with both the apicomplexan plastid and the plastid in peridinin-pigmented dinoflagellates, pointing to their common origin (*Janouskovec et al., 2010*). *Chromera* and *Vitrella* belong to a monophyletic group, called the chrompodellids, with heterotrophic colpodellids *Alphamonas edax*, *Voromonas pontica*, and '*Colpodella*' *angusta*, which also retain non-photosynthetic plastids (*Gile and Slamovits, 2014*; *Janouškovec et al., 2015*). Plastids are nevertheless absent in *Cryptosporidium* (*Abrahamsen et al., 2004*), and they have never been recorded in other deep-branching apicomplexans (*Toso and Omoto, 2007*). Lack of plastids in these groups once fueled alternative ideas about independent gains of plastids in apicomplexans, dinoflagellates, and even *Chromera* (*Bodyɫ, 2005*; *Bodyɫ et al., 2009*). However, it also brings into question the metabolic importance of the plastid for the apicomplexan cell and its implications for plastid maintenance and loss in eukaryotes more broadly (*Cavalier-Smith, 2013*; *Janouškovec et al., 2015*).

Here, we sought to understand the phylogeny and plastid evolution of Apicomplexa by filling in major gaps in their sequence data. We used individual cells of parasites to generate transcriptomes from all major apicomplexan groups that currently lack laboratory cultures and large-scale

transcriptomic or genomic data. By resolving their phylogeny, we observe that parasites with api-complexan-like morphologies are polyphyletic and originated at least three times independently. We also show that gregarines and eugregarines are monophyletic, and blastogregarines are related to archigregarines, highlighting the importance of several traits uniquely shared among them. Many deep-branching apicomplexans contain plastidial metabolism and divergent, AT-rich plastid genomes, but eugregarines lost plastids at least twice independently. Phylogeny of 16S ribosomal RNA genes and structural novelties in plastid proteins demonstrate that plastids are widespread and ancestral in the group.

# Results

## First multiprotein dataset including all major apicomplexan lineages

We generated 13 transcriptomes for 10 parasites, representing six deep apicomplexan lineages with poor presence of sequence data: protococcidians, agamococcidians, blastogregarines, archigregar-ines, eugregarines (three different superfamilies), and *incertae sedis* species (*Supplementary file 1*). Between 1 and 80 cells per species were isolated from the intestines of marine annelids, molluscs, and barnacles. The parasite cells were washed and preserved for RNA extraction (Materials and methods). Transcriptomes were sequenced from amplified cDNA by pair-end Illumina HiSeq and assembled in Trinity. To resolve deep apicomplexans relationships, we modified a published dataset of slow-evolving nucleus-encoded markers (*Derelle et al., 2016*) by including broad, representative sampling of genomes and transcriptomes of apicomplexans and related taxa (*Supplementary file 2*). This produced a phylogenetic matrix of 296 protein sequences, which were individually verified for orthology by maximum likelihood phylogenies – this allowed us to unambiguously identify paralo-gous and contaminant sequences (Materials and methods). Our apicomplexan transcriptomes typi-cally contained a single ortholog per gene; two exceptions to this were suggestive of cryptic species among the collected cells. In *Rhytidocystis* sp. 1, the most complete of multiple isoforms was selected, whereas two distinct sequence variants were present in *Siedleckia nematoides* (*Figure 1—figure supplement 1A*) and eventually merged into a single taxonomic unit (*Figure 1A*). Three other taxa were merged in the final dataset due to poor sequence representation: *Ascogregarina*, and two unidentified parasites of hexapods (*Borner and Burmester, 2017*) (*Figure 1A* and *Figure 1—figure supplement 1A*). All three are members of the superfamily Actinocephaloidea based on a consensus of protein and ribosomal RNA gene (rDNA) phylogenies (Materials and methods). The final phyloge-netic matrix contained 50 species, 99908 amino acid positions, and relatively little (10.6%) missing information (*Figure 1—source data 1*).

## Monophyly of gregarines and eugregarines, and polyphyly of apicomplexan parasites

Maximum likelihood analysis with the LG+C60+F+G4+PMSF model in IQ-TREE (non-parametric and UFBoot2 supports) and PhyloBayes analysis with the CAT+GTR model (ten independent runs) pro-duced congruent topologies that were fully resolved at most internal branches (*Figure 1A*). The pro-tococcidian *Eleutheroschizon duboscqi* was a sister of eucoccidians, and rhytidocystids branched as basal coccidiomorphs, the group containing most medically important apicomplexans. The blastog-regarine *Siedleckia* was strongly related to the archigregarine *Selenidium pygospionis*. Gregarine apicomplexans (sensu lato including blastogregarines but excluding *Digyalum*; see below) and eugregarines were both monophyletic, in contrast to most interpretations based on ribosomal RNA genes (rDNA) (*Cavalier-Smith, 2014*; *Leander, 2008*; *Rueckert et al., 2011*). To test relationships between major apicomplexan lineages, we analyzed datasets in which the two longest branches in the tree, *Cephaloidophora* and *Gregarina*, were excluded either individually or both together (LG +C60+F+G4+PMSF model with UFBoot2 supports). The resulting trees had congruent topologies with all internal branches fully supported except for two eugregarine subclades and the position of cryptosporidians (*Figure 1—figure supplement 1B*). Similarly, seven statistical tests on 105 tree topologies representing all possible relationships between coccidiomorphs, cryptosporidians, eugre-garines, archigregarines, and blastogregarines rejected all alternative topologies at p=0.01 except those differing in the placement of cryptosporidians (*Supplementary file 3*). The relationship of cryptosporidians therefore requires additional support, although their sister position to gregarines,

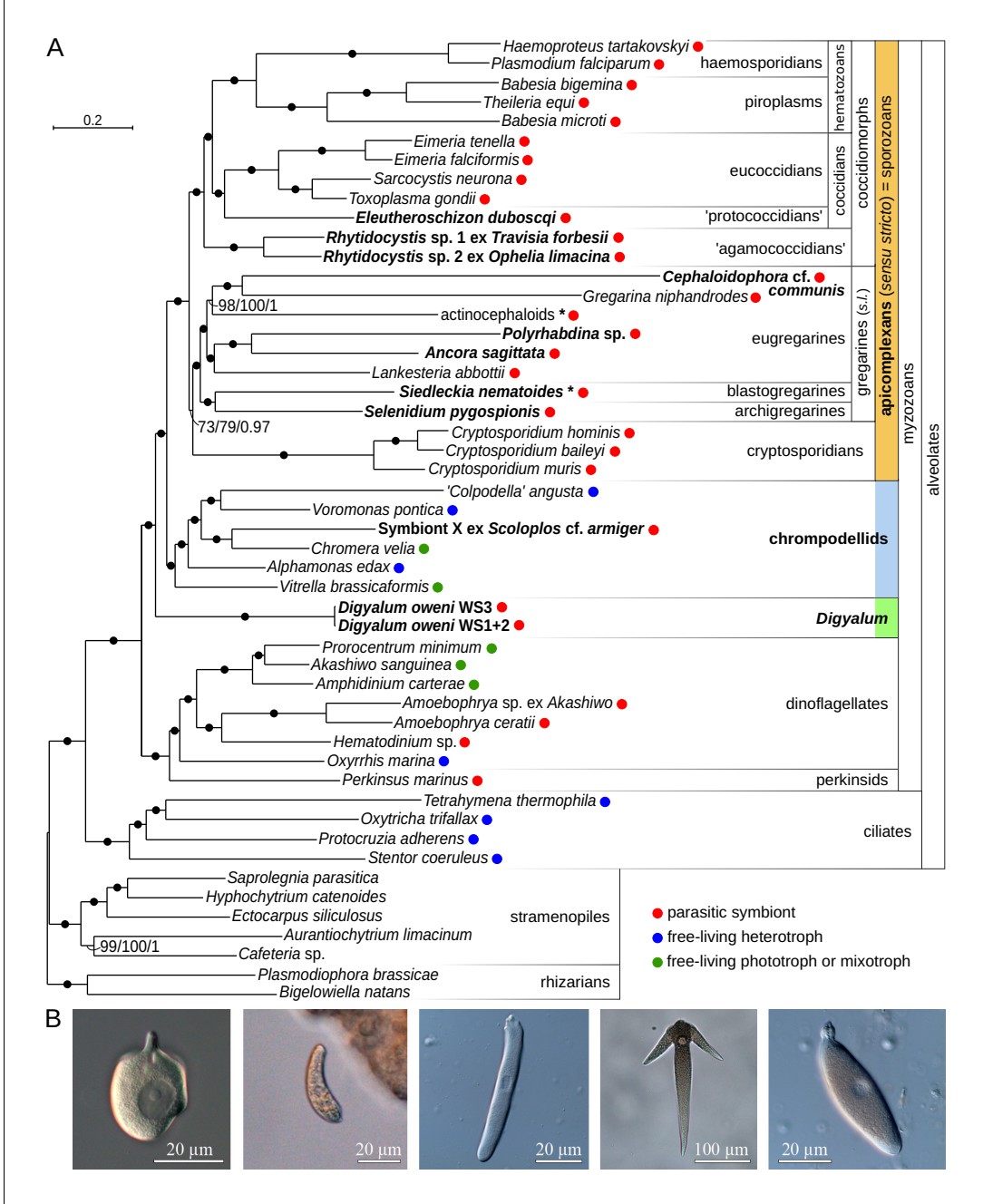

**Figure 1.** Multiprotein phylogeny of apicomplexans and related taxa. (**A**) Maximum likelihood tree (IQ-TREE) of apicomplexans and their relatives based on 296 concatenated protein markers. Species newly sequenced in this study are in bold. Values at branches correspond to UFBoot2 supports (1000 replicates, LG+G4+F+C60+PMSF model), non-parametric bootstraps (100 replicates, LG+G4+F+C60+PMSF model), and Bayesian posterior probabilities (PhyloBayes, consensus of 10 independent runs, CAT+GTR+G4 model). Black dots indicate 100/100/1 support. Actinocephaloids and *Siedleckia nematoides* are hybrid taxa (* symbol) composed from sequences of three parasites and two distant sequence variants, respectively (see *Figure 1—figure supplement 1*). Values in parentheses behind species names show % of missing data in the phylogenetic matrix. Sequence sources and the phylogenetic matrix are found in *Supplementary file 2* and *Figure 1—source data 1*, respectively. Single quotation marks indicate potentially problematic taxonomic assignments (formal group names are in *Figure 1—figure supplement 1*). (**B**) Light micrographs of some species studied, left to right: *Digyalum oweni*, Symbiont X, *Selenidium pygospionis*, *Ancora sagittata*, *Polyrhabdina* sp., with the anterior end facing up.

DOI: https://doi.org/10.7554/eLife.49662.003

The following source data and figure supplement are available for figure 1:

**Source data 1.** Phylogenetic matrix used in *Figure 1A* analysis in FASTA format.

DOI: https://doi.org/10.7554/eLife.49662.005

*Figure 1 continued on next page*

*Figure 1 continued*

**Figure supplement 1.** Multiprotein phylogenies of apicomplexans and related taxa.
DOI: https://doi.org/10.7554/eLife.49662.004

which was unambiguously recovered in all trees including the ten PhyloBayes runs, is a preferred hypothesis. Two apicomplexan-like parasites branched outside the main apicomplexan clade (*Figure 1A,B*). *Digyalum oweni*, a formally described archigregarine was fully resolved as a sister lineage to all apicomplexans and chrompodellids. A previously undescribed parasitic symbiont of the annelid *Scoloplos* with apicomplexan-like traits, named 'Symbiont X', was specifically related to *Chromera velia*. Thus, apicomplexan parasites in the traditional sense are polyphyletic (see Discussion). We keep using the name 'Apicomplexa' for the clade of parasites sensu stricto hereinafter (*Figure 1A*).

## Plastids in *Digyalum* and deep-branching apicomplexans, and their multiple losses

We next explored the existence of plastids in gregarines, *Digyalum*, and other parasites, where none have been known. Searching their sequence data with sequences of known plastid-localized proteins (*Janouškovec et al., 2015*; *Ralph et al., 2004*; *Seeber and Soldati-Favre, 2010*) revealed broad presence of plastidial pathways (Materials and methods; *Supplementary file 4*). *Digyalum*, *Selenidium, Siedleckia*, rhytidocystids, and *Eleutheroschizon* all contain near-complete plastidial biosynthesis of isoprenoid precursors, heme, and fatty acids, ferredoxin redox system, iron-sulfur cluster synthesis, and plastid genomes (*Figure 2A*). The eugregarine *Lankesteria* unusually contains fatty acid biosynthesis as the only plastidial pathway, whereas Symbiont X contains only the isoprenoid pathway, similar to piroplasms (*Lizundia et al., 2009*). Both *Lankesteria* and Symbiont X appear to lack plastid genomes (*Figure 2A*). The distribution of control, signature plastid genes involved in polypeptide import, folding, and DNA replication in the plastid (*cpn60, sDer-1, PREX*), matches the presence of plastid metabolism and genomes (*Figure 2A*). Maximum likelihood phylogenies of all individual proteins allowed us to readily distinguish the apicomplexan sequences from bacteria and other contaminants in the datasets (Materials and methods). In most phylogenies, the apicomplexan sequences cluster with algal plastid forms, confirming that they came from the plastid endosymbiont rather than the eukaryotic host. The phylogeny is different in several genes that are either derived by horizontal gene transfer from bacteria or in fact localize outside of the plastid in *Plasmodium* (in heme biosynthesis; see below). N-terminal regions of plastid sequences from our new transcriptomes often carry signal peptides typical for targeting to the plastid (the other have incomplete N-termini or lack targeting signatures by default, such as most triose phosphate translocators). The signal peptides are often followed by transit peptide-like regions, although these are more difficult to predict computationally (*Supplementary file 5*). In *Digyalum*, transit peptides have a net positive charge similar to other plastid leaders (they are low in acidic and high in basic residues; *Figure 2—figure supplement 1A*). *Digyalum* transit peptides are compositionally similar to transit peptides in *Plasmodium*, and likewise lack the phenylalanine motif at the first position after the signal peptide cleavage site (*Figure 2—figure supplement 1B*) (*Patron and Waller, 2007*). Predicted localizations for plastidial proteins and mitochondrial ALAS correspond closely to experimental evidence in *Plasmodium* and *Toxoplasma* (*Figure 2A*). The only exceptions to this pattern are the last three enzymes in heme biosynthesis, which are predicted to be plastidial in some parasites (see Discussion). The reconstructed plastid pathways also reflect known dependencies between their modules (*Figure 2B*). Iron-sulphur cluster assembly and ferredoxin system are widely required as co-factors for isoprenoid and fatty acid synthesis, whereas heme biosynthesis can be lost independently of other modules – likely the case in *Lankesteria* and Symbiont X. The synthesis of 3-phosphoglycerate (GAPDH-II and PGK-II) is present selectively. SufA was not identified in the *Chromera* genome. Pyruvate dehydrogenase and fatty and lipoic acid synthesis protein sequences in *Digyalum* are notably divergent, including three that are more closely related to bacterial than plastid sequences (*Figure 2A*). The cytosolic mevalonate pathway for isoprenoid precursor synthesis is absent in all species, but the mitochondrial cysteine desulphurase (IscS) and cytosolic fatty acid synthase (FASI) and elongase (ELO) pathways are present, as expected (Materials and methods) (*Dellibovi-Ragheb et al., 2013*;

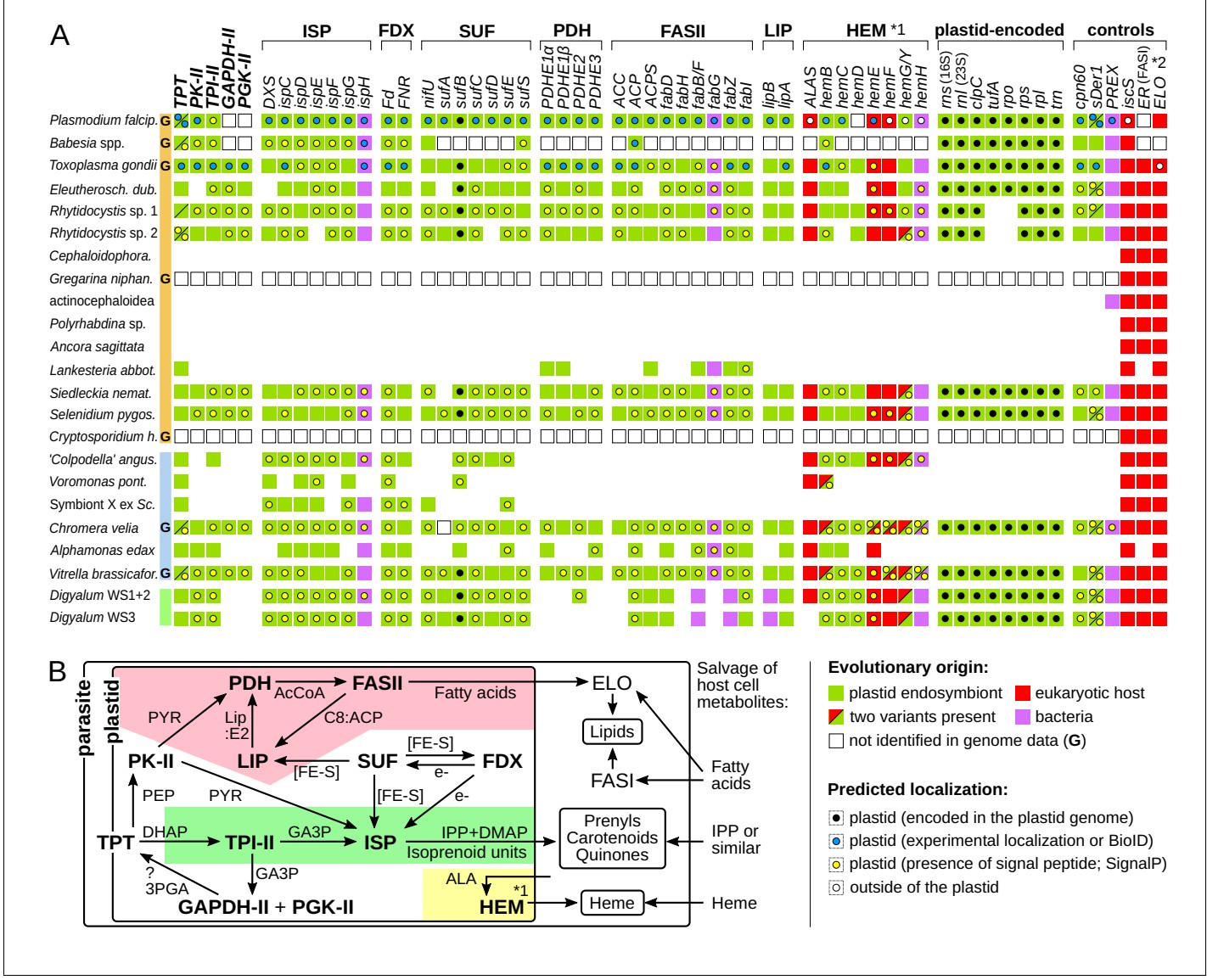

**Figure 2.** Core plastid metabolism in apicomplexans and their relatives. (**A**) Presence of genes and pathway modules (top; abbreviations in *Supplementary file 4*) in representative genomes (G) and transcriptomes (left. Each gene (box) is color-coded as to its evolutionary origin, as determined by a maximum likelihood phylogeny (plastid-encoded *rps*, *rpl*, *rpo*, *trn* genes were not analyzed). Empty boxes indicate gene absence in completed genomes and blank spaces indicate absence in transcriptomes. Intracellular localization of corresponding proteins is shown by a circle inside the box and summarizes known experimental data (*Supplementary file 4*) or de novo prediction in silico by SignalP v4.1 (*Supplementary file 5*); it is missing in proteins with incomplete N-termini. Note that only some enzymes of the heme pathway (HEM) are localized in the plastid (*1) and that signal peptides in FAS:ER and ELO were not predicted (*2). (**B**) Dependence network of plastid protein modules for the biosynthesis of key metabolites – isoprenoid precursors IPP and DMAP, fatty acids and heme – which underlie dependency on the plastid organelle in Apicomplexa. Colored regions contain modules specific to one pathway: fatty acid (pink), isoprenoid precursor (pale green) and heme biosynthesis (yellow). Interactions are reconstructed from the literature and substrates are shown near arrows (PYR = pyruvate; AcCoA = acetyl coenzyme A, Lip:E2 = lipoylation on PDHE2; C8:ACP = octanoyl:acyl carrier protein; [FE-S]=iron sulphur cluster; PEP = phosphoenolpyruvate; e-=electron reductive power; GA3p=glyceraldehyde-3-phosphate; 3PGA = 3 phosphoglycerate; ALA = δ-aminolevulinic acid; ?=uncertainty).
DOI: https://doi.org/10.7554/eLife.49662.006

The following figure supplements are available for figure 2:

**Figure supplement 1.** Transit peptides in *Digyalum* plastid proteins have positive charge but lack a conserved phenylalanine in the first position.
DOI: https://doi.org/10.7554/eLife.49662.007

**Figure supplement 2.** Maximum likelihood phylogeny of the triose phosphate translocator, TPT (LG+F+R7 model in IQ-TREE with 10000 UFBoot2 replicates;≥80 are shown; black dots indicate full support).

*Figure 2 continued on next page*

*Figure 2 continued*

DOI: https://doi.org/10.7554/eLife.49662.008

*Ramakrishnan et al., 2012*; *Zhu et al., 2004*). No plastid genes were identified in five eugregarine lineages other than *Lankesteria*, including in the draft genome of *Gregarina* (*Figure 2A*; see Discussion about the PREX fragment in *Ascogregarina*). This result is suggestive of at least two losses of plastids in eugregarines, which were independent of the one in *Cryptosporidium* (Discussion).

## Apicomplexan plastids are widespread, and their genomes are highly divergent

The discovery of plastids with genomes in deep-branching apicomplexans raises questions about whether they correspond to the undescribed apicomplexan-related lineages (ARLs) as defined by plastidial 16S rDNA (*Janouškovec et al., 2012*). Here, we discovered ten novel ARLs among environmental 16S rDNAs in GenBank and clustered 16S rDNAs obtained from the VAMPS database (*Huse et al., 2014*) by a phylogenetic sorting approach similar to the one used previously (*Janouškovec et al., 2012*) (Materials and methods). We also discontinue the use of two ARL-X and ARL-XI described recently (*Mathur et al., 2018*), which we find to be members of the *Vitrella* clade (ARL-I; *Figure 3A*) and bacterial contaminants, respectively (Materials and methods). This brings the total number of unidentified ARLs to 15, in addition to three ARLs represented by *Chromera*, *Vitrella*, and corallicolid clades (see *Supplementary file 6* for reference sequences for all ARLs). A global, maximum likelihood phylogeny of 16S rDNAs of bacteria, plastids and ARLs (including representative sequences of known ARLs and all GenBank and VAMPS centroid sequences of novel ARLs) readily illustrates that many plastids in apicomplexans and their relatives are yet to be discovered (*Figure 3A*). It also shows that some new ARLs (ARL-XV, ARL-XVI, ARL-XVII) are comparatively diversified and abundant.

Despite that ARLs have now more than doubled in number, none of them corresponds to the plastid 16S rDNAs of *Digyalum*, *Eleutheroschizon*, *Siedleckia*, *Selenidium* or *Rhytidocystis*. Instead, the five genera have some of the fastest-evolving and most AT-rich 16S rDNAs of all apicomplexans, and they cluster with compositionally similar sequences of the more distantly related hematozoans (compare *Figure 1A* and *Figure 3A*). Such artificial grouping of highly divergent sequences is a well-known phylogenetic artifact, and deep-level relationships in the tree thus ought to be interpreted with caution. The 16S rDNAs of *Digyalum*, *Eleutheroschizon*, *Siedleckia*, *Selenidium* or *Rhytidocystis* were recovered among a set of AT-rich transcriptomic contigs, which encode other genes typical of apicomplexan plastid genomes (*Supplementary file 7*; Materials and methods). In *Rhytidocystis* species 1 and 2, the AT content of 16S rDNAs (84% and 86%, respectively) and plastidial transcripts (88% and 91%, respectively) is among the highest of all plastid genomes described to date (*Figure 3B*) (*Su et al., 2019*). *Digyalum*, *Siedleckia*, *Selenidium*, and *Rhytidocystis* sp. one use a noncanonical genetic code in which UGA encodes for tryptophan; UGA is absent in the fragmentary plastid DNA of *Rhytidocystis* sp. two and encodes for STOP codons in *Eleutheroschizon* (*Figure 3B*). The *Digyalum* plastid encodes six genes never identified in apicomplexan plastids and lacks nine genes they do contain; only one gene (*rps18*) was relocated to the nucleus in parallel in both lineages (*Figure 3C*).

## Horizontal transfer, fusion and fission events in plastid-associated genes

To test if the plastids in *Digyalum*, chrompodellids and apicomplexans are likely derived from a common source we searched for shared innovations in their plastid genes. We observed that the unusual plastid DNA replication and repair complex (PREX) (*Seow et al., 2005*) is found not only in apicomplexans and chrompodellids (*Janouškovec et al., 2015*) but also in *Digyalum* (*Figure 2A* and *Figure 4*). PREX protein contains N-terminal primase and helicase domains, which are homologous to the mitochondrial primase-helicase Twinkle and fused with an *Aquifex*-type exonuclease-polymerase downstream. Phylogeny of the polymerase unit confirms that it was acquired from an unknown bacterial source related to *Aquifex* before the *Digyalum*-apicomplexan split (*Figure 4A,C*). The gene subsequently fused with the Twinkle gene and the product became targeted to the plastid (the

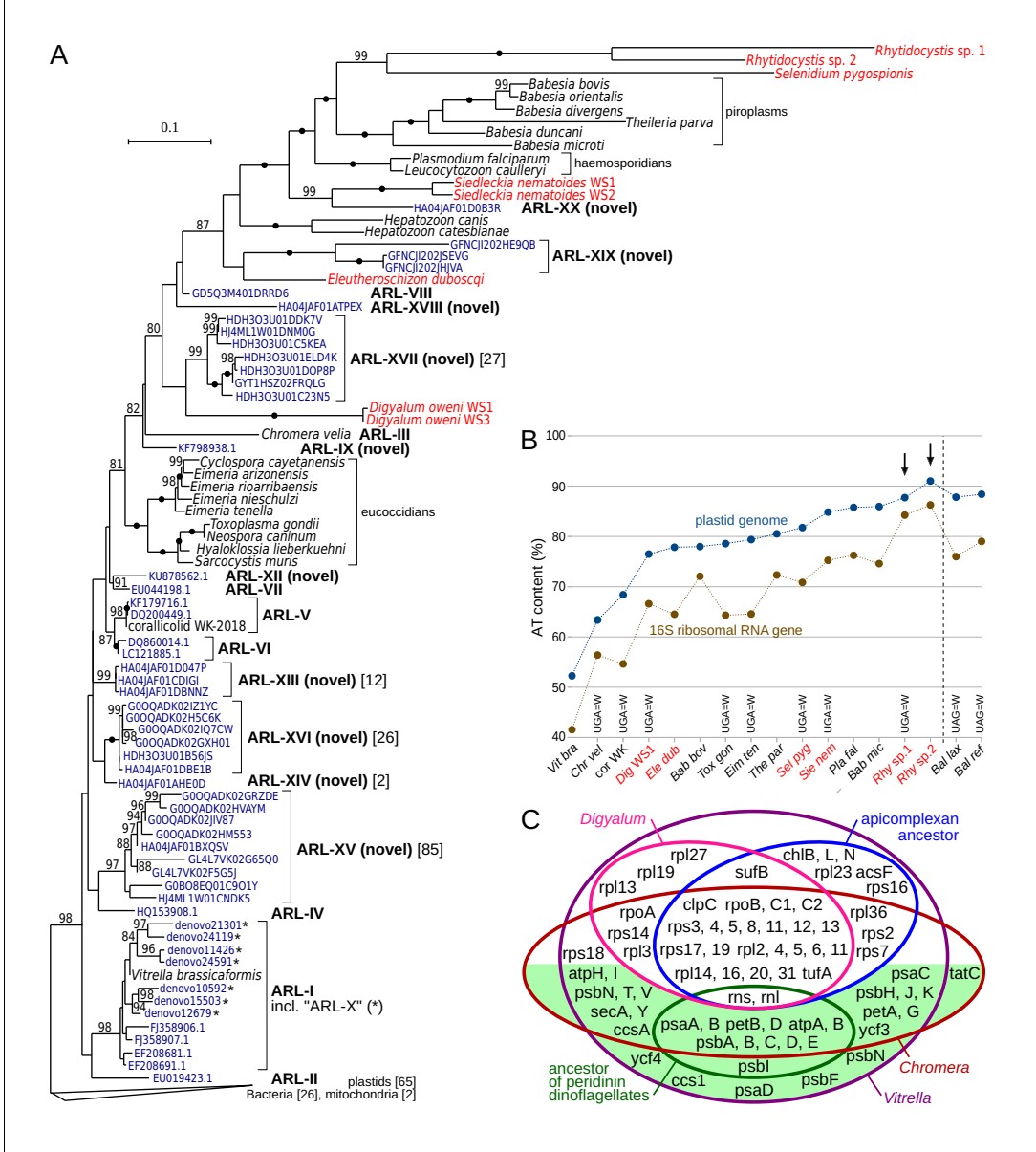

**Figure 3.** Plastid genomes in apicomplexans and their relatives are widespread and highly divergent. (**A**) Maximum likelihood phylogeny of plastid-encoded 16S ribosomal RNA genes (rDNA) reveals 10 novel apicomplexan-related lineages (ARL-IX and ARL-XII to ARL-XX; *Supplementary file 6*). Tree was computed with the best-fit TVMe+R5 model in IQ-TREE with UFBoot2 supports at branches (10000 replicates;≥80 are shown; black dots indicate 100 support). Environmental sequences (dark blue) are derived from GenBank or VAMPS (97% identity cluster centroids; numbers of reads are shown in square brackets where > 1). Plastid 16S rDNA transcripts of newly sequenced species are shown in red. Note that sequences in the tree vary greatly in their AT content and substitution rates, which can induce a misleading topology - deep relationships in the tree should therefore be interpreted with caution. The fast-evolving sequences of peridinin dinoflagellates were not included. (**B**) Extremely high AT content in rhytidocystid plastid genomes (arrows). AT content of representative species from part A and *Balanophora laxiflora* and *B. reflexa* parasitic plants (*Su et al., 2019*), all abbreviated to first three letters, is shown for 16S rDNA and plastid genomes. Plastid genomes in the newly sequenced species are only partially reconstructed from transcripts (red color; see *Supplementary file 7*). Altered genetic codes are indicated. (**C**) Euler diagram of plastid genome contents in apicomplexans, dinoflagellates (ancestral gene sets for each), *Digyalum*, *Chromera*, and *Vitrella*. Genes on the green background are associated solely with photosynthesis. Small RNA genes are not shown.

DOI: https://doi.org/10.7554/eLife.49662.009

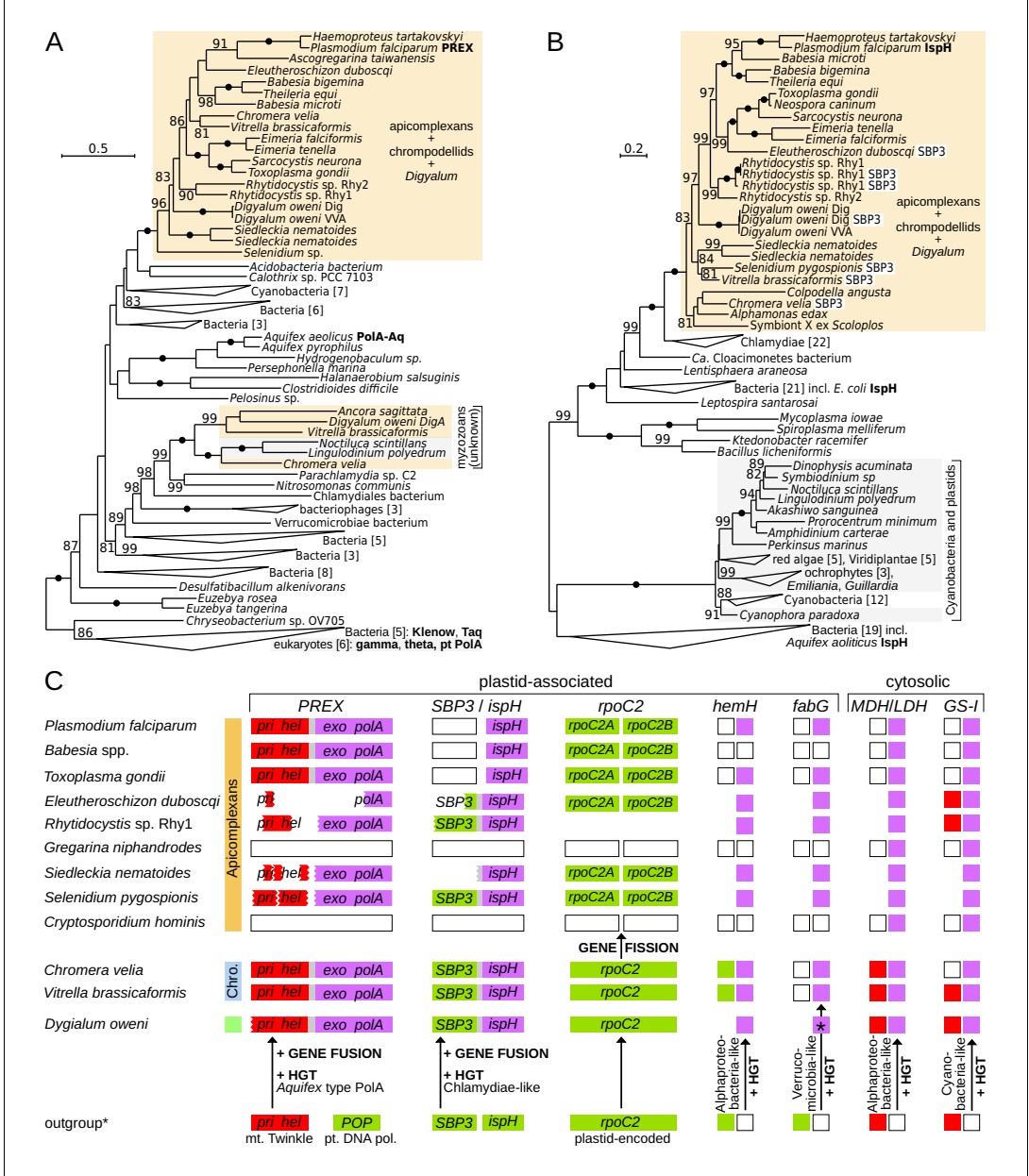

**Figure 4.** Innovations in plastid and cytosolic genes define early evolution of apicomplexans and their relatives. (A) Maximum likelihood phylogeny of the exonuclease/polymerase subunit of the plastid replication and repair complex, PREX (LG+R6 model). (B) Maximum likelihood phylogeny of 4-hydroxy-3-methylbut-2-enyl diphosphate reductase, IspH (LG+I+G4 model). Trees were derived from protein sequences in IQ-TREE and have UFBoot2 supports at branches (10000 replicates; ≥80 are shown; black dots indicate 100 support). Apicomplexans, chrompodellids and *Digyalum* are highlighted in orange, and other eukaryotes in gray. Characterized enzymes are highlighted in bold. Fusion *ispH* genes with *SBP3* at the N-terminus are shown in white boxes. (C) Predicted gain, loss, fusion and fission events in the evolution of five plastid-associated and two cytosolic genes. Genes are shown by boxes (jagged edges indicate truncated genes) and are color-coded by origin as in *Figure 2A* (plastid endosymbiont = green, eukaryotic host = red, bacteria = purple, absent in genome data = white, absence of evidence = blank area). Outgroup (*) shows the state in the closest relevant comparator: dinoflagellates (IspH, HemH, MDH/LDH, GS-I) or other algae (PREX, SBP3, RpoC2, FabG). Note that the ferrochelatase (HemH) is mitochondrial in *Plasmodium* but probably plastidial in *Chromera* and some apicomplexans. Abbreviations: POP = plant organellar DNA polymerase, Twinkle = mitochondrial primase/helicase, SBP3 = sedoheptulose-1,7-bisphosphatase form 3, MDH/LDH = malate/lactate dehydrogenase (other abbreviations in *Supplementary file 3*).

DOI: https://doi.org/10.7554/eLife.49662.010

The following figure supplements are available for figure 4:

**Figure supplement 1.** The region of the split in the apicomplexan plastid-encoded *rpoC2*.

*Figure 4 continued on next page*

*Figure 4 continued*

DOI: https://doi.org/10.7554/eLife.49662.011

**Figure supplement 2.** Maximum likelihood phylogeny of the ferrochelatase, HemH (LG+R8 model in IQ-TREE with 10000 UFBoot2 replicates;≥80 are shown).

DOI: https://doi.org/10.7554/eLife.49662.012

**Figure supplement 3.** Maximum likelihood phylogeny of the beta-ketoacyl-acyl carrier protein reductase, FabG (LG+R6 model in IQ-TREE with 10000 UFBoot2 replicates;≥80 are shown).

DOI: https://doi.org/10.7554/eLife.49662.013

N-terminus of PREX is incomplete in most species including *Digyalum* but contains a signal peptide in *Chromera*; **Supplementary file 5**). Another unusual fusion took place in 4-hydroxy-3-methylbut-2-enyl diphosphate reductase (IspH), the last enzyme in the plastid isoprenoid precursor biosynthesis. A canonical plastid *ispH* gene of a cyanobacterial origin, which is also present in dinoflagellates and *Perkinsus*, was replaced by a Chlamydiae-like variant in the ancestor of *Digyalum* and Apicomplexa and fused with the plastid gene for sedoheptulose-1,7-bisphosphatase form 3 (SBP3) (**Figure 4B,C**). The fusion is absent in *Toxoplasma*, *Plasmodium* and piroplasms, which lack *SBP3*, and has been interpreted as a derived characteristic in *Chromera* (**Petersen et al., 2014**), but instead we find it is broadly distributed in apicomplexans and their relatives.

Another unique evolutionary event in apicomplexan plastids is a fission in a non-conserved region of the plastid-encoded *rpoC2* gene. The unprecedented split was once interpreted as a read-through frame shift (in *Plasmodium*) or read-through STOP codons (in *Toxoplasma* and *Eimeria*) in a continuous *rpoC2* (**Cai et al., 2003**; **Wilson et al., 1996**). We instead find that all apicomplexan *rpoC2* genes are split within the same region, including those in the deep-branching *Selenidium* and *Siedleckia* (**Figure 4C** and **Figure 4—figure supplement 1**). The two *rpoC2* moieties are found in different reading frames in most species, but two different in-frame STOP codons (UAA and UAG) separate them in *Toxoplasma*. Both frame shifting and STOP codon read-through would be required for continuous *rpoC2* translation in *Eleutheroschizon* (**Figure 4—figure supplement 1**). Such variable gene arrangements are incongruent with the expression of the apicomplexan *rpoC2* as a single protein. Messenger RNA editing likewise does not correct the *rpoC2* reading frame in *Plasmodium* (**Nisbet et al., 2016**). Indeed, the downstream *rpoC2* moiety almost unequivocally possesses an ATG start codon near the split site allowing it to be translated independently (**Figure 4—figure supplement 1**). The evidence altogether points to a fission event in *rpoC2*, which is absent in all other plastids, including those of *Digyalum* and chrompodellids, and thus represents a defining ancestral characteristic of the apicomplexan plastid. Two additional plastid-associated proteins in apicomplexan and chrompodellid plastids derive from horizontally acquired genes. The first is Alphaproteobacteria-like ferrochelatase (HemH), which is localized to mitochondria in *Plasmodium* but likely targeted to plastids in *Chromera* and possibly in some apicomplexans (see Discussion) (**Koreny et al., 2011**; **Sato and Wilson, 2003**; **Varadharajan et al., 2004**), and the other is Verrucomicrobia-like beta-ketoacyl-acyl carrier protein reductase (FabG) (**Janouškovec et al., 2015**). *Digyalum* encodes the same alphaproteobacterial HemH but it has an unusual FabG, which is related to other bacteria (**Figure 4C**, **Figure 4—figure supplement 2** and **Figure 4—figure supplement 3**). Finally, genes of two well-known cytosolic proteins (**Huang et al., 2004**; **Zhu and Keithly, 2002**) were also acquired from bacteria in the *Digyalum*-Apicomplexa ancestor: lactate/malate dehydrogenase and glutamine synthase type I (**Figure 4C**).

## Discussion

### Convergent evolution of apicomplexan-like morphology

Generating transcriptomes from uncultured apicomplexans across their evolutionary diversity provides the first comprehensive insights into relationships between major apicomplexan groups. Two species with apicomplexan-like morphology either described as apicomplexans (*Digyalum*) or yet unclassified (Symbiont X) are not members of Apicomplexa sensu stricto. This shows that apicomplexan-like parasites are polyphyletic and evolved at least three times independently. Specifically, large trophont stages attached to intestines of marine invertebrates by specialized apical structures

are products of convergent evolution. The trophonts of *Digyalum*, for example, parasitize the gut epithelium of *Littorina* snails and their attachment structure contains an apical complex with a protruded polar ring, which provides a gateway for rhoptry-mediated secretion – a combination of traits typical for gregarine apicomplexans (*Dyson et al., 1994*; *Dyson et al., 1993*). Symbiont X is known only from light microscopy data but would be also readily classified as an apicomplexan based on crude characteristics: it parasitizes the gut of *Scoloplos armiger* polychaetes being attached to the host epithelium by its apical end (unpublished data). Tracing the evolution of such parasite characteristics, however, indicates that the basis for convergence lies in evolution acting on similar preconditions. Apical complexes with rhoptries, micronemes and pseudoconoids are present in free-living relatives of apicomplexans, such as in the predatory *Colpodella* and *Psammosa* and photosynthetic *Chromera*, and in more distantly related parasites such as *Perkinsus* and *Parvilucifera* (*Foissner and Foissner, 1984*; *Norén et al., 1999*; *Oborník et al., 2011*; *Okamoto and Keeling, 2014*; *Perkins, 1996*). Such broad distribution points to a single origin of the apical complex in the ancestor of apicomplexans and dinoflagellates in a non-parasitic context (*Figure 1A*). Because the apical complex mediates secretion often associated with cell-to-cell interactions, it is likely an important precondition in multiple origins of parasitism in both apicomplexans and dinoflagellates. In similar host environments the structure may have also promoted convergent parasite morphologies. The use of the apical complex in extracellular attachment and secretion in the gut epithelium of animal hosts, for example, may have triggered convergent expansion in the cell size of gregarine, *Digyalum* and Symbiont X trophonts. Unsurprisingly, convergent similarities in the three lineages are accompanied by considerable differences in detailed morphology: *Digyalum* and Symbiont X do not glide or twist but they pulsate, and detailed ultrastructure of their apical complex and pellicle (*Dyson et al., 1994*; *Dyson et al., 1993*) (unpublished data) is distinct from that in gregarines (*Kováčiková et al., 2017*; *Paskerova et al., 2018*; *Valigurová et al., 2017*). Similar divergence characterizes their molecular make-ups: apicomplexans are well-known auxotrophs for purines, but *Digyalum* contains a pathway for their synthesis (data not shown) and, despite that both lineages lost photosynthesis, their plastid genomes have been reduced in different ways (*Figure 3C*). The convergent morphologies of *Digyalum*, Symbiont X and apicomplexans are therefore rather superficial similarities, but the possibility that they were driven by the presence of shared ancestral traits (such as the apical complex) in similar host habitats highlights the importance of preconditions in the origin of parasites (*Janouskovec and Keeling, 2016*).

## Relationships and morphological transitions in early apicomplexan evolution

Key findings of multiprotein phylogenies are that eugregarines and gregarines are unequivocally monophyletic (*Figure 1* and *Figure 1—figure supplement 1*). This broadly supports traditional morphological classifications (*Grassé, 1953*) in contrast to many recent proposals based on rDNA phylogenies, which regard both groups as polyphyletic (*Cavalier-Smith, 2013*). Indeed, protein sequences allow for building larger phylogenetic matrices and have more even substitution rates than rDNAs, some of which are notoriously divergent and phylogenetically unstable. Although the sampling of eugregarine diversity is incomplete, our phylogeny contains six of their seven main lineages at the superfamily level (*Simdyanov et al., 2017*) – one of them being *Polyrhabdina*, which is apparently not a lecudinoid (*Figure 1A* and unpublished data). The eugregarine monophyly provides the first unambiguous phylogenetic support for their two candidate synapomorphies: the ultrastructure of the epimerite, and the ultrastructure of epicytic crests (*Simdyanov et al., 2017*). It also partially resolves the little understood relationships between eugregarine superfamilies (*Cavalier-Smith, 2014*; *Simdyanov et al., 2017*; *Simdyanov et al., 2015*). The blastogregarine *Siedleckia* groups strongly with the archigregarine *Selenidium*. The sampling of both groups is incomplete, especially in the archigregarines, but they do share a combination of characteristics that are rare or absent in other apicomplexans, namely active bending and twisting movement in trophozoites, pellicular folds running along the body length in many species, and one or more layers of longitudinal microtubules underlying the pellicle (*Schrével et al., 2016*; *Simdyanov et al., 2018*). Both lineages also feed by myzocytosis, which is known in some relatives of apicomplexans (*Foissner and Foissner, 1984*; *Mylnikov and Mylnikova, 2008*) but not in eugregarines, cryptosporidians or coccidiomorphs. Because the blastogregarine ultrastructure and life cycle share additional similarities with coccidians (*Simdyanov et al., 2018*), confirming their phylogenetic position is important for

reconstructing character evolution in Apicomplexa as a whole. The monophyly of gregarines sensu lato in our trees (*Figure 1* and *Figure 1—figure supplement 1*) provides phylogenetic support for a tentative synapomorphy of this group, the forming of a gametocyst during the life cycle (*Simdyanov et al., 2017*), although this characteristic is absent in blastogregarines. The position of cryptosporidians is still not fully resolved, but the group is consistently recovered next to gregarines (*Figure 1* and *Figure 1—figure supplement 1*). *Eleutheroschizon* is clearly important to understanding the origin of eucoccidians (*Figure 1A*) (*Valigurová et al., 2015*), although its classification as a protococcidian is apparently incorrect (unpublished data). Two *Rhytidocystis* species are positioned as a basal group in the class Coccidiomorpha (*Figure 1A*), which supports traditional views on their classification closer to coccidians (*Levine, 1979*; *Porchet Hennere, 1972*) rather than to gregarines (*Cavalier-Smith, 2014*). The bigger group that includes rhytidocystids and related parasites of annelids and molluscs (*Kristmundsson et al., 2011*) therefore provides a stepping stone for understanding the evolution of coccidiomorphs, including the majority of medically important apicomplexans.

## Widespread maintenance but multiple losses of plastids in apicomplexa

Evidence for endosymbiotic genes with plastid targeting signals and plastid genomes shows that plastids are common in deep-branching apicomplexans, despite not having been previously recognized (*Figure 2*, *Figure 3* and *Supplementary file 5*). Their plastid metabolic networks are similar to those in *Toxoplasma* and *Plasmodium* and generally serve to produce three essential metabolites: isoprenoid precursors for the synthesis of carotenoids, ubiquinone and prenyls; heme as a protein co-factor; and fatty acids as constituents of cellular lipids (*Imlay and Odom, 2014*; *Ramakrishnan et al., 2012*; *van Dooren et al., 2012*). Heme biosynthesis is partially localized in the cytosol and mitochondria, and fatty acids can be extended by non-plastidial elongases (*Figure 2A*) (*Ramakrishnan et al., 2012*; *Zhu et al., 2004*), but de novo biosynthesis of all three metabolites outside of the plastid appears to be absent. This creates dependence on the plastid for the production of isoprenoid precursors, heme, and fatty acids, which can be overcome only by metabolite uptake from the environment. Salvage of host isoprenoids, heme, and fatty acids is common in parasites, but obtaining them in sufficient amounts across the life cycle is apparently challenging. Most parasites retain all three plastidial pathways, and only two lineages have lost plastid organelles altogether (*Figure 5*): cryptosporidians (*Abrahamsen et al., 2004*), and the dinoflagellates *Hematodinium* and *Amoebophrya* (*Gornik et al., 2015*; *John et al., 2019*). In free-living organisms, plastids are always retained, perhaps because replacing their metabolism by salvage is even more difficult (*Janouškovec et al., 2017*). Most consistently required is the biosynthesis of isoprenoid precursors (*Figure 5*), which is the only indispensable plastidial pathway in piroplasms, the *Plasmodium* blood stage, and possibly in Symbiont X (*Figure 2A*) (*Lizundia et al., 2009*; *Yeh and DeRisi, 2011*). A fully unprecedented situation exists in the eugregarine *Lankesteria*, where fatty acid biosynthesis appears to be the only retained plastidial anabolic pathway (the transcriptome is fragmentary but enzymes for the biosynthesis of isoprenoid precursors and heme are absent). Interestingly, the existence of plastids in *Selenidium* and *Rhytidocystis* is also consistent with ultrastructural reports of multimembrane organelles in some of their species, which correspond to non-photosynthetic plastids in other apicomplexans by size, appearance, and their position near the nucleus (*Figure 5*) (*Leander and Ramey, 2006*; *Porchet Hennere, 1972*; *Schrével, 1971*; *Schrével et al., 2016*; *Wakeman et al., 2014*). Five eugregarines lack any evidence for plastid presence; these include *Gregarina* where a draft genome is available (*Figure 2A*). Since *Lankesteria* is phylogenetically nested among them, eugregarines must have lost plastids at least twice (*Figure 5*). *Ascogregarina* contains a fragment of *PREX* (*Figure 2A* and *Figure 4A*), but because the species lacks evidence for other endosymbiont-derived genes we conservatively consider it encodes for an ex-plastidial protein or is a contaminant. Our results indicate that the two plastid losses in eugregarines are independent of the one in cryptosporidians (*Figure 1A*), unlike previous assumptions (*Cavalier-Smith, 2014*; *Toso and Omoto, 2007*). This increases total known losses of plastid organelles in apicomplexans to three and in all eukaryotes to four (*Figure 5*).

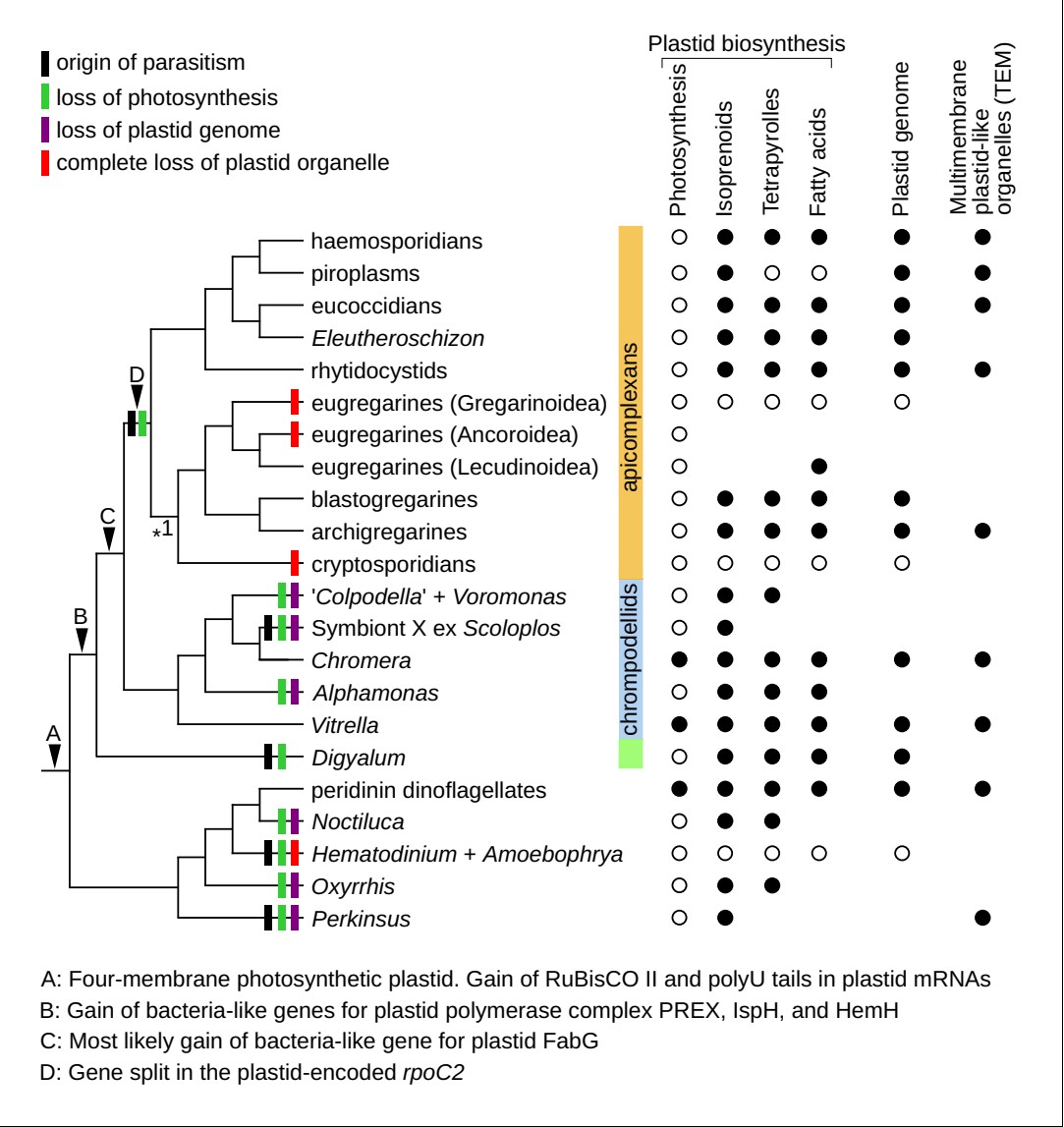

**Figure 5.** Plastid evolution in apicomplexans and their relatives. Plastid-related characteristics (**A–D**), origin of parasitism, and predicted losses of photosynthesis, plastid genomes, and plastid organelles were mapped on the updated phylogeny. The phylogeny is fully resolved except one branch where more support is needed (*1). Predicted core plastid anabolic capabilities, presence of plastid genomes, and transmission electron microscopy evidence (TEM) for multimembrane organelles corresponding to plastids by their size, appearance, and position within cells are shown on the right. DOI: https://doi.org/10.7554/eLife.49662.014

## Plastids in apicomplexans and their relatives were inherited vertically from a common ancestor

Plastids in *Chromera* and *Vitrella* share several unique characteristics with the apicomplexan plastid (*Janouskovec et al., 2010*) but their common origin was doubted in the past (*Bodył et al., 2009*) and other early endosymbiotic events could underlie plastid evolution in the broader group including dinoflagellates (*Waller and Kořený, 2017*). Testing whether plastids in apicomplexans, chrompodellids and *Digyalum* derive from the same endosymbiont is therefore relevant to understanding plastid evolution and frequency of plastid losses in general. We find three lines of evidence that support a common origin. Firstly, phylogenies of individual plastid genes repeatedly group *Digyalum* with apicomplexans and chrompodellids, and their topologies either reiterate nuclear phylogenies directly or are unresolved without consistently supporting alternatives (*Figure 4A,B*, *Figure 4—figure supplement 2* and *Figure 4—figure supplement 3*). Secondly, unidentified lineages that carry

plastid-encoded 16S rDNAs related to apicomplexans and chrompodellids (ARLs) are growing in number and diversity (*Figure 3A*). Some are apicomplexans (*Janouškovec et al., 2013*; *Kwong et al., 2019*) and others may be free-living, but they altogether support the idea that plastids in the lineage are widespread (*Janouškovec et al., 2012*). Finally, unique evolutionary innovations in the plastidial *PREX*, *ispH*, and *hemH* link together the plastids in apicomplexans, chrompodellids, and *Digyalum* (bacterial *fabG* and split *rpoC2* further link plastids in some of them). It is difficult to see how endosymbiosis would move plastids in or out of the lineage without changing the distribution of these genes. Altogether, the evidence suggests that plastids in apicomplexans, chrompodellids, and *Digyalum* were inherited vertically from a common ancestor, are widely distributed in the group, and are most likely retained by default, particularly in free-living representatives (*Janouškovec et al., 2015*; *Janouskovec et al., 2010*).

## Metabolic evolution in the plastid

Core plastid metabolism in *Digyalum*, Symbiont X, and apicomplexans has been remarkably conserved across long time scales and reveals only a few outstanding variations. In *Plasmodium falciparum,* triose phosphate sugars are imported by two triose phosphate translocators (TPTs) residing in the outermost (PfoTPT) and innermost (PfiTPT) plastid membranes, respectively (*Mullin et al., 2006*). *Toxoplasma* contains only one TPT phylogenetically corresponding to PfoTPT. Orthologs of the PfoTPT lack N-terminal targeting signatures and are ubiquitous in apicomplexans, chrompodellids, and *Digyalum* (*Figure 2—figure supplement 2*). PfiTPT orthologs are less common (*Gile and Slamovits, 2014*), but they possess N-terminal signal peptides in *Chromera*, *Vitrella*, and *Rhytidocystis* sp. two compatible with their targeting to the inner plastid membrane (*Figure 2A* and *Figure 2—figure supplement 2*). The split between the two forms therefore likely predated Apicomplexa and the loss of the PfiTPT form in the lineage leading to *Toxoplasma* is apparently a derived evolutionary state. One enzyme that processes the TPT substrate dihydroxyacetone phosphate (DHAP) is triose phosphate isomerase (TPI-II), and the failure to identify TPI-II in piroplasms once led to the proposition that their plastids import glyceraldehyde-3-phosphate, the TPI-II product (*Fleige et al., 2010*). We find that TPI-II, although highly divergent in piroplasms, is present in all apicomplexans with plastids, and it frequently possesses N-terminal signal peptides for plastid targeting (*Figure 2A*). This suggests that the import and conversion of DHAP is conserved across apicomplexan plastids.

Analysis of heme biosynthesis enzymes suggests that the pathway consistently starts in the mitochondrion, as in heterotrophic eukaryotes – the algal C5-pathway must have been lost prior to the *Digyalum*-Apicomplexa divergence. Delta-aminolevulinic acid is likely imported to the plastid and processed to coproporphyrinogen III (*Ralph et al., 2004*; *Sato et al., 2004*). The last three enzymes in some apicomplexans sequenced by us are unexpectedly predicted to be also plastidial, similarly to *Chromera* and *Vitrella* (*Koreny et al., 2011*). In *Plasmodium*, and perhaps also in *Toxoplasma* and *Digyalum*, the last three heme biosynthesis enzymes localize in the cytosol and mitochondria, where heme is most needed (*Varadharajan et al., 2004*). It would be interesting to explore these contrasting localization predictions experimentally, including the possibility that some enzymes are dually targeted, or their isoforms (where present; *Figure 2A*) are differentially targeted to different cellular compartments.

## Highly divergent plastid genomes and their multiple losses in chrompodellids

Unlike metabolism in the plastid, plastid genome structure shows unexpected variations across Apicomplexa. Plastid genomes in some of the newly sequenced taxa, as partially reconstructed from transcripts, have unusually AT-rich and fast-evolving sequences (*Figure 3A*). Completing the fragmentary plastid genome of *Rhytidocystis* species 2 (*Supplementary file 7*) is of particular interest because it has the most AT-rich 16S rDNA, and potentially the most AT-rich genome, ever recorded among plastids (*Figure 3B*) (*Su et al., 2019*). Plastid genome reduction in *Digyalum* and Apicomplexa from their common ancestor is primarily explained by the loss of photosynthesis (*Figure 3C*). This was accompanied by relatively modest transfer of genes to the nucleus, which involved remarkably different gene sets in the two lineages (only *rps18* was relocated in both in parallel).

Underlying the very existence of plastid DNA in *Plasmodium* and *Toxoplasma* is *sufB*, which encodes one of only two broadly conserved, plastid-encoded proteins with function other than

transcribing and translating the genome itself. Of the newly sequenced species with plastids, only Symbiont X lacks any evidence for the *sufB* gene or plastid genome (*Figure 3A*). It would be expected that Symbiont X *sufB* is encoded in the nucleus and was relocated there in its common ancestor with *Chromera*, *Voromonas*, *Colpodella*, and *Alphamonas* (*Janouškovec et al., 2015*), thus allowing heterotrophs in this lineage to lose plastid genomes at least three times independently (*Figure 5*).

## Summary and future directions

We provide a strongly resolved phylogeny and large-scale sequence data for major apicomplexan groups, but the sampling of Apicomplexa is far from complete. Expanding the phylogenetic dataset with new species and genes will allow for testing key conclusions of our study (e.g., monophyly of gregarines and eugregarines) and understanding the relationships of taxa that remain poorly sampled (archigregarines, blastogregarines, protococcidians) or lack genome-level sequences (corralicolids, adeleid and aggregatid coccidians, and other *incertae sedis* taxa). Parasites with characteristics similar to apicomplexans could provide a study system for morphological and molecular convergence and insights into the transition from free-living species to obligate symbionts. Plastids in apicomplexans and their relatives are apparently ancestral and widespread, and more are likely to be discovered. The plastid function in apicomplexans, chrompodellids and *Digyalum* rarely includes photosynthesis, but it always involves synthesis of one or more indispensable metabolites (*Figure 5*). Losses of plastid organelles and their genomes are infrequent but did occur several times in the broader group, and are likely to provide an unparalleled model for understanding factors that mediate plastid maintenance and loss in eukaryotes as a whole.

Interestingly, two key conclusions of our study are independently reinforced by a bioRxiv preprint (*Mathur et al., 2019*), which describes a complementary set of parasite transcriptomes from the same group. Firstly, the manuscript reports two parasites with apicomplexan-like morphologies that likewise branch outside Apicomplexa. The early-branching *Platyproteum*, a representative of 'squirmids' (*Cavalier-Smith, 2014*), is the sister group of *Digyalum*, as has been apparent to us from 18S rDNA phylogenies (unpublished data). *Piridium* then represents a sister taxon to *Vitrella* and therefore a fourth independent emergence of apicomplexan-like parasitism in the lineage. Secondly, plastids appear to be consistently absent in most eugregarine superfamilies except for the Lecudinoidea (*Lankesteria* in our study and *Lecudina* and *Pterospora* in the other study), where fatty acid biosynthesis is the only core plastidial pathway. The presence of two additional eugregarine superfamilies in our trees (Ancoroidea and *Polyrhabdina*) points to an extra case of plastid loss (*Figure 5*), but the relationships of eugregarine superfamilies which are present in both studies are fully congruent. Integrating sequence datasets from both studies will be a first step in creating a phylogenetic framework for apicomplexan evolution. This framework will likely be useful in illuminating steps in the emergence of parasitism and in predicting cell biology of less known parasites by the methods of comparative genomics.

## Materials and methods

**Key resources table**

| Reagent type (species) or resource | Designation | Source or reference | Identifiers | Additional information |
|---|---|---|---|---|
| Commercial assay or kit | RNAqueous-Micro Total RNA Isolation Kit | ThermoFisher | cat no. AM1931 | |
| Commercial assay or kit | SMART-Seq v4 Ultra Low Input RNA Kit | Takara | cat no. 634888 | |
| Software, algorithm | MAFFT v7.402 | *Katoh and Standley, 2013* | RRID:SCR_011811 | |

*Continued on next page*

Continued

| Reagent type (species) or resource | Designation | Source or reference | Identifiers | Additional information |
|---|---|---|---|---|
| Software, algorithm | IQ-TREE v1.6.5 | *Nguyen et al., 2015* | RRID:SCR_017254 | |
| Software, algorithm | PhyloBayes v4.1c (and MPI v1.7b) | *Lartillot et al., 2009* | RRID:SCR_006402 | |

## Parasite sampling, transcriptome sequencing and assembly

Parasite cells (1 to approximately 70 individuals) were isolated from marine annelid, mollusc and barnacle hosts collected at the White Sea Biological Station of Moscow State University during August of 2016 (*Supplementary file 1*). Cells were hand-picked by using a glass micropipette (30% ethanol was used to detach the cells of *Digyalum* and *Eleutheroschizon*), washed 1 to 4 times in clean seawater and transferred into a clean tube in the final volume of 2–5 ul of seawater (excess seawater removed if necessary after a brief spin). Care was taken to avoid contamination by animal host cells and gut contents when isolating parasite cells but we were not able to fully prevent it (host contaminants were observed in the sequence data but they were clearly distinguishable from the parasites; see below). A 100 ul of Lysis buffer (RNAqueous-Micro Total RNA Isolation Kit; Ambion/Thermo-Fisher, cat no. AM1931) was added into the tube with parasite cells and the samples were stored at −80C for several weeks. Total RNA was extracted from samples by RNAqueous-Micro Total RNA Isolation Kit according to manufacturer instructions but without the DNase I digest step. In *Digyalum* and *Eleutheroschizon*, RNA from two independent cell isolations was combined before further processing (*Supplementary file 1*). Cells of *Digyalum oweni* WS3-2017 was stored in RNALater and used for reverse-transcription without extracting RNA. RNA was reverse-transcribed by using SMART-Seq v4 Ultra Low Input RNA Kit (Takara; cat no. 634888), however20-30 amplification cycles; optimization was done as in SMARTer Pico PCR cDNA Synthesis Kit). Indexed TruSeq libraries were built at Edinburgh Genomics and sequenced as paired-end 150 bp reads in two multiplexed lanes on the Illumina HiSeq 4000 machine. For the *Digyalum oweni* WS3-2017 sample, paired-end 150 bp HiSeq 4000 reads were produced independently of other samples. Demultiplexed reads were processed by cutadapt v1.8.3 to remove low quality nucleotides (-q 20 setting), polyA tails, SMARTer adapters and leftover Illumina barcodes (on 3' ends of shorter sequence fragments). Reads of minimum length of 100 nucleotides were assembled in Trinity v2.4.0 by using the default settings. Predicted proteomes were generated as six-frame translations of transcriptomic contigs. Analysis of assembled reads revealed minor cross-contamination between samples due to adapter swapping, but contaminants were well distinguishable by read coverage. Raw sequence reads and our assemblies were deposited in NCBI Bioprojects PRJNA557242 and PRJNA556465. Photographs of studied parasites (*Figure 1B*) were made using Leica DM 2500 microscopes equipped with DIC optics and a digital camera (Leica, Germany) at the White Sea Biological Station of Moscow State University and the Marine Biological Station of St. Petersburg State University.

## Single protein sequence alignments for nuclear phylogeny

We built our dataset from 339 protein sequence alignments previously used in stramenopile phylogenies and tested for orthology (*Derelle et al., 2016*). Firstly, slow-evolving stramenopile sequence query for each alignment (mostly the *Phytophthora* sequence) was used to retrieve five best BLASTP hits (e-value cutoff of 1e-5) and select the closest ortholog in translated alveolate, stramenopile, and rhizarian genomes (*Supplementary file 2*). Secondly, a new alveolate protein sequence query (primarily from *Vitrella*) was used in the same way to expand the dataset with our newly generated and existing transcriptomes. Closest hits from animal, fungal, microsporidian and bodonid genomes were also included to distinguish sequences derived from animal host and other contaminants in the samples. Orthologous sequences were identified by multiple rounds of alignment clean-up as guided by Maximum Likelihood phylogenies. Initial rounds of phylogenies were focused on removing out-paralogs and contaminants by using default alignment in MAFFT v7.402 (*Katoh and Standley, 2013*), alignment trimming in BMGE v1.12 (*Criscuolo and Gribaldo, 2010*) with the -b 3 g 0.4

setting and phylogeny in Fasttree v2.1.10 (*Price et al., 2010*) with the -lg -gamma setting. Later rounds were focused on resolving difficult cases of paralogy, horizontal gene transfer, and the orthology of fast-evolving sequences: localpair, –linsi, alignment in MAFFT; -b 4 g 0.4 settings in BMGE; and IQ-TREE v1.6.5 (*Nguyen et al., 2015*) phylogeny with the LG+I+G4+F model and 1000 UFBoot2 supports. Of the original 339 single protein sequence alignments (see *Supplementary file 1* in *Derelle et al., 2016*), 43 were excluded in the process: 10 were absent in apicomplexans and dinoflagellates (alignments #72, #290, #291, #292, #294, #295, #300, #301, #304 and #315) and 33 were sparsely sampled or strongly incongruent with known organismal relationships (alignments #29, #57, #70, #73, #82, #149, #151, #155, #157, #159, #163, #164, #167, #171, #183, #184, #191, #193, #194, #198, #241, #246, #260, #265, #270, #283, #285, #289, #302, #303, #305, #330, #331). In the remaining 296 alignments, paralogs were reduced to the most slowly evolving sequence with non-conflicting phylogenetic placement (else both were removed) and multiple gene isoforms were reduced to the most complete sequence. Cross-contaminant sequences were identified based on read coverage and excluded. Divergent regions of sequences resulting from frame shift errors or imperfect gene models were removed from alignments manually. Adjacent protein sequence fragments that were apparently derived from the same gene fragmented by wrong genome annotation were merged.

## Multiprotein phylogeny and topology tests

The final 296 protein sequence datasets were realigned by the localpair algorithm in MAFFT, trimmed by using the -b 4 g 0.4 settings in BMGE and concatenated in Scafos v1.25 (*Roure et al., 2007*). During the latter step, sequences derived from different strains of the same species (in *Colpodella angusta*, *Sarcocystis neurona*, *Voromonas pontica*, and unidentified actinocephaloid parasites of *Helicoverpa armigera* and *Helicoverpa assulta*) were merged into single operational taxonomic units (*Supplementary file 2*). The initial phylogenetic matrix contained 54 species, 99948 sites and 14.4% missing data (*Figure 1—figure supplement 1A*). To further reduce missing data we merged sequences of two *Oxyrrhis marina* strains and of two distinct variants found in *Siedleckia nematoides* transcriptomes; the latter are possibly derived from different cryptic species (variant one was preferentially identified and/or found to be more complete in the transcriptome of the WS1 strain; variant two in the transcriptome of the WS2 strain). We also merged sequences of three representatives of the superfamily Actinocephaloidea with low sequence presence: *Ascogregarina* and two unidentified parasites of the insects *Teleopsis* and *Helicoverpa*, which contaminate the host transcriptomes (*Borner and Burmester, 2017*). The 18S rDNA of the *Teleopsis* parasite branches among other actinocephaloids (data not shown) and both unidentified group with *Ascogregarina* in the multiprotein phylogeny (*Figure 1—figure supplement 1A*). Merging all these taxa produced the main phylogenetic matrix with 50 species, 99908 positions and 10.6% missing data (*Figure 1—source data 1*). Three additional matrices were created by excluding all sequences of *Gregarina*, *Cephaloidophora*, or both species (*Figure 1—figure supplement 1B*). Final matrices were first analyzed in IQ-TREE by using the LG+I+G4+F model and the best tree was used as a guide tree in a more thorough analysis with the LG+G4+F+C60+PMSF model: 1000 UFBoot2 replicates were computed for all datasets (*Figure 1A* and *Figure 1—figure supplement 1*) and 100 non-parametric bootstrap were computed for the main matrix (*Figure 1A*). Seven statistical tests of 105 tree topologies corresponding to all possible relationships between coccidiomorphs, cryptosporidians, eugregarines, archigregarines, and blastogregarines in *Figure 1A* were calculated in IQ-TREE v1.6.5 with the LG+I+G4+F model and 10000 replicates (*Supplementary file 3*). The main phylogenetic matrix (*Figure 1A*) was also analyzed by 10 independent PhyloBayes runs (either standard version 4.1 c or MPI version 1.7b) (*Lartillot et al., 2009*) with the GTR+CAT model and constant sites removed (-dc setting). Each chain was run for 1000 cycles, of which the initial 250 were discarded and the remaining (10 × 750) were combined to compute a consensus tree (maxdiff = 0.23, meandiff = 0.00057).

## Analysis of plastidial metabolism and horizontally acquired genes

We next searched the orthologs of apicomplexan plastid proteins with focus on core pathways (*Figure 2A*) in transcriptomes and genomes of apicomplexans and their relatives (including *Perkinsus* and dinoflagellates) by using similar approach as for the nuclear genes above. Five best BLASTP hits at the e-value threshold of 1e-5 were retrieved for each species by using a comparatively slow

evolving sequence query (typically *Vitrella* or *Chromera*). These hits were included in plastid protein alignments created previously (*Janouškovec et al., 2017*; *Janouškovec et al., 2015*), or in new alignments created here by retrieving representative outgroup sequences from GenBank and the local database by BLASTP searches with the same query sequences (*Supplementary file 4*). This process included non-plastidial genes of the heme biosynthesis pathway, and three genes representing controls for non-plastidial pathways: *iscS* (mitochondrial iron-sulfur biosynthesis),enoyl reductase domain (FASI, cytosolic fatty acid synthesis), and *ELO* (endoplasmic reticulum-localized fatty acid elongation). Alignments were reduced by an iteration process analogous to that used for nuclear genes. Final phylogenetic matrices were prepared by localpair alignment in MAFFT and -b 4 g 0.4 trimming in BMGE, and analyzed in IQ-TREE by using the built-in ModelFinder to select the best model with the preset LG substitution matrix (for example, LG+F+R7 model for TPT in *Figure 2— figure supplement 2*). This approach allowed to distinguish true apicomplexan sequences from contaminants in our transcriptomes and confirmed their origins to be in the plastid endosymbiont (grouping with other plastidial sequences) or eukaryotic host (grouping with eukaryotic cytosolic or mitochondrial sequences; *Figure 2A*). Protein phylogenies of six horizontally acquired genes (*Figure 4A,B*, *Figure 4—figure supplement 2* and *Figure 4—figure supplement 3*) were built by expanding previous datasets (*Janouškovec et al., 2015*) and computed by using the same method as plastid phylogenies above.

## Analysis of leader sequences in plastid proteins

Proteins of core plastidial pathways (*Figure 2A*) derived from our transcriptomes and those of *Plasmodium falciparum*, *Babesia microti* and *Toxoplasma gondii* were scanned for the presence of identifiable N-terminal signal peptides in SignalP v4.1 (*Petersen et al., 2011*). All methionines downstream of the predicted protein start were also tested for targeting signals. Positive sequences were checked for the presence of N-terminal extensions – truncated proteins were filtered out as false positives. The remaining signal peptide-positive proteins were recorded in *Figure 2A* and further checked for identifiable transit peptides in ChloroP 1.1 (*Emanuelsson et al., 1999*). The SignalP and ChloroP statistics were listed in *Supplementary file 5*. Apicomplexan proteins that have known experimental localization (as described in primary literature and Apiloc3) or that were classified as high-confidence plastid-localized proteins in *Plasmodium falciparum* by BioID (*Boucher et al., 2018*), were recorded in *Figure 2A* and listed in *Supplementary file 4*. To characterize transit peptides in *Digyalum oweni*, we expanded its identified plastid protein set. Additional plastidial proteins in the WS1+two isolates were searched by BLASTP with apicomplexan queries, comprising known *Toxoplasma* and *Plasmodium* plastid proteins primarily involved in transcript and protein processing (e.g., stromal processing peptidase, clpP chaperone, histone-like protein HU, tRNA-Met ligase, etc.). Matched sequences were verified by reverse BLASTP searches on KEGG and NCBI BLAST websites (aided by distance tree phylogenies at the latter site). Pooling positive hits with those identified previously (*Figure 2A*) and filtering for sequences with N-terminal extension carrying a signal peptide (as identified by SignalP v4.1) produced 41 plastid proteins. The first 14 transit peptide residues downstream of the signal peptide cleavage site (an approach to allow comparison with results in *Patron and Waller, 2007* were merged and analyzed altogether for their amino acid composition by 'Protein Stats' script at the Sequence Manipulation Suite website. Composition of the 41 mature proteins (without transit peptides) was also analyzed – because transit peptide cleavage sites are difficult to predict in silico the initial 50 residues were removed arbitrarily (*Figure 2—figure supplement 1A*). The amino acid frequency at the first 20 positions across the 41 *Digyalum* transit peptides was analyzed at the WebLogo3 website (*Figure 2—figure supplement 1B*).

## Plastid genome analysis and 16S rDNA phylogeny

Plastids transcripts were identified by three search strategies: BLASTN searches with plastidial 16S rDNA; BLASTP searches with sequences of plastid-encoded protein sequences of apicomplexans or chrompodellids;and searches for high AT contigs (typically contigs > 70% AT and 1–3 kb in length). Hits from the former search were examined in a 16S rDNA tree. Hits those from the latter two search strategies were combined and reversely compared by BLASTX against the NCBI nr database to search if they match genes in apicomplexan, chrompodellids and algal plastid genomes. Positive hits were limited to representative, non-redundant contigs of 1 kb or longer (*Supplementary file 7*).

Identification of individual genes was based on NCBI BLAST searches and tRNAscan-SE, and it was further aided by Artemis 17.0.1 and MFannot website. AT content was examined in 16S rDNAs and in whole plastid genomes or combined plastid transcripts by the 'DNA Stats' script on the Sequence Manipulation Suite website (*Figure 3B* and *Supplementary file 7*). The phylogeny of 16S rDNA was based on an earlier dataset with representatives of ARL-I to ARL-VIII clades (*Janouškovec et al., 2012*). We also requested sequences of ARL-X and ARL-XI as reported in *Mathur et al. (2018)* from the authors of the study. Of the 34 sequences of clustered centroids we received, only one centroid was named 'ARL-XI' and this was a *Pelagibacter*-like bacterial contaminant; we therefore consider ARL-XI to be invalid. The remaining 33 centroids were named 'ARL-X': nine of them were bacterial contaminants and the remaining formed two non-overlapping groups of 3 and 21 sequences. We included in our phylogeny all three sequences from the first group and four slowly evolving sequences from the second group, but they all branched within ARL-I (*Figure 3A*), which leads us to synonymize ARL-X with ARL-I. Finally, we included in the phylogeny 10 new ARLs identified by phylogenetic sorting of the VAMPS database (*Huse et al., 2014*). Briefly, we obtained all VAMPS reads annotated as 'Organelle' or 'Unknown', selected those being 350 bp or longer, and clustered each group in Usearch at 97% identity. Centroids were individually classified by maximum likelihood phylogenies in a dataset containing plastids, bacteria and mitochondria by an approach used previously (*Janouškovec et al., 2012*). The initial round of phylogenies was computed by using Fasttree2 (-lg -gamma setting) and later rounds by using IQ-TREE (LG+I+G4+F model; additional rounds with modified taxon sampling were used for sequences that were difficult to classify). Candidate ARL sequences were used to retrieve additional ARLs from centroid databases by BLASTN searches (i.e., those that had been misplaced by Fasttree2). The apicomplexan affiliation of all ARL sequences was verified in the *Figure 1A* dataset. Representative sequences for all known and novel ARLs were listed in *Supplementary file 6*. The final 16S rDNA phylogeny was based on a localpair MAFFT alignment trimmed in BMGE (-h 0.4 g 0.65 settings) and computed by using IQ-TREE with Model-Finder selection of the best fit model and 10000 UFBoot2 supports (*Figure 3A*).

## Acknowledgements

We thank Ross F Waller and reviewers and editors in *Elife* for providing helpful comments and suggestions about the draft manuscript. We thank the staff of the White Sea Biological Station of Lomonosov Moscow State University and the Marine Biological Station of St. Petersburg State University for assistance during field sampling. The authors acknowledge the use of the UCL Myriad High Performance Computing Facility (Myriad@UCL), and associated support services, in the completion of this work. *Digyalum oweni* WS3 transcriptome sequencing was funded by the Russian Science Foundation (grant no. 18-14-00123).

## Additional information

### Funding

| Funder | Grant reference number | Author |
|---|---|---|
| University College London | Excellence Research Fellowship | Jan Janouškovec |
| Russian Foundation for Basic Research | 18-04-00324 | Gita G Paskerova<br>Timur G Simdyanov<br>Tatiana S Miroliubova |
| Russian Science Foundation | 18-14-00123 | Vladimir V Aleoshin |
| Saint Petersburg State University | 1.42.1099.2016 | Gita G Paskerova |
| Saint Petersburg State University | 1.42.723.2017 | Gita G Paskerova |

The funders had no role in study design, data collection and interpretation, or the decision to submit the work for publication.

## Author contributions
Jan Janouškovec, Conceptualization, Data curation, Formal analysis, Supervision, Funding acquisition, Validation, Investigation, Visualization, Methodology, Writing—original draft, Project administration, Writing—review and editing; Gita G Paskerova, Timur G Simdyanov, Funding acquisition, Investigation, Methodology, Writing—review and editing; Tatiana S Miroliubova, Investigation, Methodology, Writing—review and editing; Kirill V Mikhailov, Formal analysis, Methodology, Writing—review and editing; Thomas Birley, Formal analysis, Visualization, Writing—review and editing; Vladimir V Aleoshin, Supervision, Funding acquisition, Methodology, Writing—review and editing

## Author ORCIDs
Jan Janouškovec (iD) https://orcid.org/0000-0001-6547-749X
Gita G Paskerova (iD) https://orcid.org/0000-0002-1026-4216
Vladimir V Aleoshin (iD) http://orcid.org/0000-0002-3299-9950
Timur G Simdyanov (iD) http://orcid.org/0000-0003-2478-9301

## Decision letter and Author response
Decision letter https://doi.org/10.7554/eLife.49662.028
Author response https://doi.org/10.7554/eLife.49662.029

# Additional files

## Supplementary files
• Supplementary file 1. Information about parasites collected (single quotation marks indicate problematic affiliations; asterisk denotes a species complex).
DOI: https://doi.org/10.7554/eLife.49662.015

• Supplementary file 2. Sources of sequence datasets used in this study (single quotation marks indicate arbitrarily assigned strain name).
DOI: https://doi.org/10.7554/eLife.49662.016

• Supplementary file 3. Results of seven tree topology tests as computed in IQ-TREE (of 105 tested topologies those not rejected at p=0.001 in at least one test are listed; those not rejected at p=0.01 are highlighted in bold).
DOI: https://doi.org/10.7554/eLife.49662.017

• Supplementary file 4. Names, abbreviations, accessions and localizations of plastidial proteins and modules (plastid-localized = green; mitochondrion-localized = blue; endoplasmic reticulum = red).
DOI: https://doi.org/10.7554/eLife.49662.018

• Supplementary file 5. Prediction statistics for signal (SignalP v4.1: noTM, Dmaxcut = 0.45) and transit peptides (ChloroP v1.1).
DOI: https://doi.org/10.7554/eLife.49662.019

• Supplementary file 6. Reference 16S rDNA sequences for apicomplexan related lineages (ARLs), *Digyalum*, and newly sequenced apicomplexans.
DOI: https://doi.org/10.7554/eLife.49662.020

• Supplementary file 7. Plastid transcriptomic contigs in species newly sequenced in this study.
DOI: https://doi.org/10.7554/eLife.49662.021

• Transparent reporting form
DOI: https://doi.org/10.7554/eLife.49662.022

## Data availability
Sequence data have been deposited in NCBI under the Bioproject accessions PRJNA557242 and PRJNA556465. Sources of data for individual analyses are provided in Supplemental Tables S1 to S7.

The following datasets were generated:

| Author(s) | Year | Dataset title | Dataset URL | Database and Identifier |
|---|---|---|---|---|
| Janouskovec J, Paskerova GG, Symdianov TG | 2019 | Transcriptomes of apicomplexan parasites | https://www.ncbi.nlm.nih.gov/bioproject/?term=PRJNA557242 | NCBI Bioproject, PRJNA557242 |
| Miroliubova TS, Mikhailov KV, Aleoshin VV | 2019 | Transcriptome of Digyalum oweni | https://www.ncbi.nlm.nih.gov/bioproject/PRJNA556465 | NCBI Bioproject, PRJNA556465 |

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
