## [Decision Letter]

Thank you for submitting your article "Convergent origins of parasitism, new phylogenetic relationships and complex distribution of plastids in Apicomplexa." for consideration by *eLife*. Your article has been reviewed by three peer reviewers, and the evaluation has been overseen by John McCutcheon as the Reviewing Editor and Detlef Weigel as the Senior Editor. The following individuals involved in review of your submission have agreed to reveal their identity: Christopher Howe (Reviewer #1); Geoff McFadden (Reviewer #3).

The reviewers have discussed the reviews with one another and the Reviewing Editor has drafted this decision to help you prepare a revised submission.

Summary:

The phylum Apicomplexa includes many medically important and biologically fascinating organisms. However, only a few species of medical importance have been studied with genomics, leaving many important questions related to the evolution of their unusual non-photosynthetic plastids and their sometimes parasitic lifestyles unanswered. Using transcriptome sequencing from apicomplexan cells isolated from various marine invertebrates, the authors resolved some of the long-standing mysteries of Apicomplexa. They show that parasitism likely evolved not once but many times in Apicomplexa, and they show that plastid loss is likely more common that originally appreciated. The evolution of this important group of organisms is now much clearer.

Essential revisions:

There are five essential revisions that need to be made.

1) The difficulty of reconstructing the deeper regions of the phylogenies (because of biased sequence compositions, rates of evolution) is mentioned and dealt with thoroughly but needs to be made more clear in the text for readers that are not experts in phylogenetic reconstructions.

2) The potential problems associated with bacterial contamination in transcriptomes should be discussed more clearly in the main text. The reviewers thought that some reporting on gene-level support for each key gene *not* being the result of contamination would be useful. If most genes follow a certain pattern (for example, if all are found on the plastids of close relatives), then the results could be summarized in a sentence or two in the Results. If the results are more variable, then perhaps a table would be more appropriate. We appreciate that some detail on this is provided in the Materials and methods, and the main line of evidence seems to be that the transcripts group with plastid-genome-encoded sequences from other species where genomes are available, but this needs to be clearly articulated in the appropriate Results section. For example, were analyses performed in which all of the sequences corresponding to the previously-reported ARLs were included, or just centroids? More detail on these analyses might help the reader to understand the issue more clearly.

3) The existence of a related bioRxiv preprint (http://dx.doi.org/10.1101/636183) which reports many of the same findings needs to be acknowledged and discussed. The reviewers and editors feel that this would strengthen the paper and is the fair thing to do. The authors needn't worry about *eLife* rejecting their manuscript because of this bioRxiv preprint (see this article on *eLife*'s policy on scooping: https://elifesciences.org/articles/30076).

4) The terminology related to Apicomplexa, apicomplexans, plastids, apicoplasts, etc. needs to be clarified and made consistent throughout the manuscript. It is clear that parasitism has arisen multiple times in this lineage. That's interesting. What is murky is whether the apical complex has arisen independently, which is what some of the language in this paper hints at (e.g Abstract, Introduction, last paragraph and subsection “Monophyly of gregarines and eugregarines, and polyphyly of apicomplexan parasites”. Where does this paper stand on parallel origins of the apical complex? Given that we have some structures in dinoflagellates that we think are homologues, when does a case of parallel evolution from a common ancestral structure become identifiable? Bat and bird wings is a nice example. Both arose independently from a common ancestral structure, namely the vertebrate forelimb. They are still homologous, but modification to aid flight arose convergently. Are the authors of this submission arguing that the apical complex arose independently multiple times, or are they simply saying that parasitism arose multiple times amongst the lineage that has Phylum Apicomplexa at its tip? There is a really big difference of course. The problem is that select lineages could also have lost or degraded apical complexes. We certainly see many parasites with stages that don't develop an apical complex. It seems that the authors are saying (at least in the title) that the apical complex arose once, and that parasitism has arisen multiple times amongst a lineage possessed of an apical complex, but the conclusion from the text is less clear.

The authors have chosen not to use the term apicoplast. Given its wide usage in the literature, a mention of the term early on, and justification for using the more general term plastid to embrace the homologous organelles in those relatives without an apicoplast complex such as the dinozoa, would be appropriate in the Introduction.

Another language issue is whether or not the authors are using formal or informal names for taxa. Happily, they adopt the systematics of Adl et al., 2018, which is widely accepted in the protistology field. The problems encountered in this submission were mainly when the informal name was capitalised (e.g. the tree in Figure 1). This is confusing and should be avoided as it hints at the name being a formal taxon. It is therefore preferable to avoid using informal taxon names at the start of sentences. Clear usage would also include the taxon rank (e.g. Phylum Apicomplexa) for formal taxon names, but lower case for informal categories (e.g. apicomplexans). It is also important to be consistent. For instance, in Figure 1 the authors use the term Cryptosporidians (the capital C should go) but in the text the frequently use the term cryptosporidia. Choose one informal name, and stick to it throughout.

5) The title needs to be made more precise and exciting.

---

## [Author Response]

Essential revisions:There are five essential revisions that need to be made.1) The difficulty of reconstructing the deeper regions of the phylogenies (because of biased sequence compositions, rates of evolution) is mentioned and dealt with thoroughly but needs to be made more clear in the text for readers that are not experts in phylogenetic reconstructions.

This is a good suggestion. To address it, we extended the description of the plastid 16S rDNA phylogeny (the bias affects exclusively the analysis in Figure 3A). We clarified that the deep-level topology needs to be interpreted with caution and explained that unrelated but compositionally similar sequences can cluster artificially together. Although none of this bias relates to our interpretations it is indeed better that readers are made aware of it (we ask them to compare trees in Figures 3A and 1A). The relevant sentence in the Figure 3 legend was also reworded to clarify that “sequences in the tree vary greatly in their AT content and substitution rates, which can induce a misleading topology – deep relationships in the tree should therefore be interpreted with caution”.

2) The potential problems associated with bacterial contamination in transcriptomes should be discussed more clearly in the main text. The reviewers thought that some reporting on gene-level support for each key gene not being the result of contamination would be useful. If most genes follow a certain pattern (for example, if all are found on the plastids of close relatives), then the results could be summarized in a sentence or two in the Results. If the results are more variable, then perhaps a table would be more appropriate. We appreciate that some detail on this is provided in the Materials and methods, and the main line of evidence seems to be that the transcripts group with plastid-genome-encoded sequences from other species where genomes are available, but this needs to be clearly articulated in the appropriate Results section. For example, were analyses performed in which all of the sequences corresponding to the previously-reported ARLs were included, or just centroids? More detail on these analyses might help the reader to understand the issue more clearly.

We resolved these issues in the following ways. In nuclear phylogenies in the Results, we highlighted that single gene orthologs were identified by “maximum likelihood phylogenies – this allowed us to unambiguously identify paralogous and contaminant sequences (Materials and methods”. In the plastid phylogeny section, we clarified how contaminants were identified and how the origin of individual genes was assigned, as requested: “Maximum likelihood phylogenies of all individual proteins allowed us to readily distinguish the apicomplexan sequences from bacteria and other contaminants in datasets (Materials and methods). In most phylogenies, the apicomplexan sequences form a cluster that is related to algal plastidial forms confirming an origin in the plastid endosymbiont rather than eukaryotic host). The phylogenetic pattern is only different in genes that are derived by horizontal gene transfer from bacteria or that in fact localize outside of the plastid in *Plasmodium* (heme biosynthesis; see below).” Finally, a description of the global ARL phylogeny with regard to selection of environmental sequences was added, as requested. We included it in Results where we briefly describe the tree and explain that it includes “newly identified VAMPS centroids and representative sequences of known ARLs”.

3) The existence of a related bioRxiv preprint (http://dx.doi.org/10.1101/636183) which reports many of the same findings needs to be acknowledged and discussed. The reviewers and editors feel that this would strengthen the paper and is the fair thing to do. The authors needn't worry about eLife rejecting their manuscript because of this bioRxiv preprint (see this article on eLife's policy on scooping: https://elifesciences.org/articles/30076).

We agree. This is an exciting report that reaches some similar conclusions based on a fully complementary sets of apicomplexan transcriptomes. We referenced the preprint in the last paragraph of Discussion entitled “Summary and future directions”. We briefly compared the principal findings of the two studies, and the agreement of their conclusions in two main areas: the polyphyly of Apicomplexa and multiple plastid losses in eugregarines. Importantly, while the two studies do not fully overlap, they do not fundamentally disagree on any particulars. We note that integrating the two datasets will provide a strong framework for understanding apicomplexan evolution.

4) The terminology related to Apicomplexa, apicomplexans, plastids, apicoplasts, etc. needs to be clarified and made consistent throughout the manuscript. It is clear that parasitism has arisen multiple times in this lineage. That's interesting. What is murky is whether the apical complex has arisen independently, which is what some of the language in this paper hints at (e.g Abstract, Introduction, last paragraph and subsection “Monophyly of gregarines and eugregarines, and polyphyly of apicomplexan parasites”. Where does this paper stand on parallel origins of the apical complex? Given that we have some structures in dinoflagellates that we think are homologues, when does a case of parallel evolution from a common ancestral structure become identifiable? Bat and bird wings is a nice example. Both arose independently from a common ancestral structure, namely the vertebrate forelimb. They are still homologous, but modification to aid flight arose convergently. Are the authors of this submission arguing that the apical complex arose independently multiple times, or are they simply saying that parasitism arose multiple times amongst the lineage that has Phylum Apicomplexa at its tip? There is a really big difference of course. The problem is that select lineages could also have lost or degraded apical complexes. We certainly see many parasites with stages that don't develop an apical complex. It seems that the authors are saying (at least in the title) that the apical complex arose once, and that parasitism has arisen multiple times amongst a lineage possessed of an apical complex, but the conclusion from the text is less clear.

This is an important point that was now clarified in the text. The evidence is clear that apicomplexan parasites are polyphyletic, but this makes no assumption about single vs. multiple origins of their apical complex organelles. The reviewer understood us correctly that the apical complex originated only once but this had not been stated explicitly. We have now expanded the text and references in corresponding section of the Discussion, among others clearly saying that the “distribution points to a single, early origin of the apical complex in the ancestor of apicomplexans and dinoflagellates, in a non-parasitic context”. We then link the early presence of the apical complex to the emergence of convergent parasite morphologies, as outlined previously, so this new information fits in very well.

The authors have chosen not to use the term apicoplast. Given its wide usage in the literature, a mention of the term early on, and justification for using the more general term plastid to embrace the homologous organelles in those relatives without an apicoplast complex such as the dinozoa, would be appropriate in the Introduction.

Yes, we now introduced the term in the Introduction. We explain that “The apicoplast is a four-membrane plastid (a broader term we will use hereinafter to describe the organelle in both parasitic and free-living organisms)”.

Another language issue is whether or not the authors are using formal or informal names for taxa. Happily, they adopt the systematics of Adl et al., 2018, which is widely accepted in the protistology field. The problems encountered in this submission were mainly when the informal name was capitalised (e.g. the tree in Figure 1). This is confusing and should be avoided as it hints at the name being a formal taxon. It is therefore preferable to avoid using informal taxon names at the start of sentences. Clear usage would also include the taxon rank (e.g. Phylum Apicomplexa) for formal taxon names, but lower case for informal categories (e.g. apicomplexans). It is also important to be consistent. For instance, in Figure 1 the authors use the term Cryptosporidians (the capital C should go) but in the text the frequently use the term cryptosporidia. Choose one informal name, and stick to it throughout.

We agree. Given the complex taxonomic history of the group, different formal names or ranks are still being used for Apicomplexa or its subgroups (e.g., phylum vs. subphylum for Apicomplexa; note that Adl et al., 2019, move away from taxonomic ranks altogether). Since the taxonomy is not in the focus here either, we have adopted informal names more widely in the text and in Figure 1A. We kept formal taxonomic group names in Figure 1—figure supplement 1 for comparison, and we point to this in the Figure 1 legend. The one formal name that we keep using more widely is “Apicomplexa”, which we now introduce as a phylum in the Introduction. We avoided instances of informal names at sentence beginnings and used single forms (“cryptosporidians”) throughout the manuscript, as suggested.

5) The title needs to be made more precise and exciting.

We rephrased the title in an active voice, which makes it more interesting and explicit at the same time: it highlights two key discoveries, which the reviewers also pointed out in their summary.